# Metagenomic Assembled Genomes of a *Pseudanabaena* Cyanobacterium and Six Heterotrophic Strains from a Xenic Culture

**DOI:** 10.3390/microorganisms13091996

**Published:** 2025-08-27

**Authors:** Paul D. Boudreau

**Affiliations:** Department of BioMolecular Sciences, University of Mississippi School of Pharmacy, Oxford, MS 38677, USA; boudreau@olemiss.edu; Tel.: +1-(662)-915-3612

**Keywords:** DNA extraction, cell lysis, microbiome, whole genome sequencing

## Abstract

Sequencing cyanobacteria from xenic cultures is often challenging when their DNA extracts are confounded by DNA from their heterotrophic microbiome. Using an iterative DNA lysis protocol can fractionate between DNA from the cyanobacterium and the heterotrophic strains. To further demonstrate the utility of this protocol, it was used to sequence another xenic culture of cyanobacteria. This effort led to the assembly of a megabase-length cyanobacterial chromosome; however, repeated ribosomal regions created assembly issues even after adding data from another sequencing run to improve coverage. A separate DNA preparation from a single cell lysis step was also run for comparison but yielded a markedly lower proportion of cyanobacterial reads (<2%). Instead, the circular cyanobacterial chromosome was closed with targeted amplicon sequencing. Phylogenetic analysis assigned this strain to the genus *Pseudanabaena*. Within the metagenomic assembly were the genomes of six heterotrophic strains, preliminarily assigned as belonging to the genera *Acidovorax*, *Hydrogenophaga*, *Lysobacter*, *Novosphingobium*, *Sediminicoccus*, and *Tabrizicola*. *Lysobacter* sp. BL-A-41-H3’s chromosome was also assembled as a closed circular contig. This study demonstrates that iterative lysis enriches for cyanobacterial DNA and enables concurrent genome assembly of cohabitating heterotrophs alongside the host cyanobacterium.

## 1. Introduction

Cyanobacteria are an ancient lineage of Gram-negative bacteria, the only organisms on Earth to have evolved oxygenic photosynthetic bacteria [1,2]. They were responsible for oxygenating the Earth’s atmosphere over two billion years ago [3] and still play central roles in the planet’s carbon and nitrogen cycles [2,4]. They are of interest to the research community for potential applications in biofuels or drug discovery, work that has been hindered by relatively few model strains being available [5]. As photosynthetic organisms, cyanobacteria need large cells (relative to other bacteria) to ensure they can capture sufficient light energy for growth, indeed, some cyanobacteria can form macroscopic assemblages of quasi-multicellular filaments [1]. This one adaptation to a photosynthetic lifestyle leads to a host of problems in terms of extracting and isolating DNA from these organisms, namely a poor ratio of DNA to cell biomass, though polyploidy has been observed in this clade [6,7]. They also have a robust cell membrane and cell wall (and, in some cases, an external rigid polysaccharide sheath), which helps these large cells maintain their structure but resists lysis [6,8], as well as the presence of smaller heterotrophic bacterial cells on the surface of cyanobacterial cells, DNA from which can confound sequencing runs [9,10]. On top of this, cyanobacteria are well known for genomic diversity and for having large genomes, meaning the genome assembly post-sequencing is a challenge relative to most other bacteria [11].

Taken together, these challenges may explain the under-representation of cyanobacterial genomes in public sequence databases [12]. Evidence for this can be seen in the Genome Taxonomy Database, which currently has 7057 cyanobacterial entries compared to 59,430 actinomycetes, another clade of highly productive bacteria for drug discovery [13,14,15]. However, the difference in cellular morphology between cyanobacteria and their heterotrophic microbiome can be appropriated using iterative lysis steps to fractionate between cyanobacterial and heterotrophic DNA, using one problem (difficult to lyse cyanobacterial cells) to solve another (confounding reads from heterotrophic strains) [16]. Importantly, high-quality large contig assemblies of their genomes are more needed than ever. Recent genome sequencing efforts have shown inconsistencies in older phylogenic classification of cyanobacteria based on morphology and amplicon sequencing, hinting that we do not have a full picture of this ancient clade [17].

Cyanobacteria are also privileged producers of natural products [5,8,18,19], and with the biosynthetic gene clusters responsible for these compounds often being large pathways, with multiple clusters dispersed across the genome, a near complete genome sequence is needed to understand the true biosynthetic capacity of a strain [11]. While tools for predicting natural product chemical structure from gene sequence are improving [20], these tools are dependent on quality genome assemblies.

Hypothesizing that the iterative lysis approach could be applied to target the DNA of other cyanobacteria in xenic cultures, not just the two strains it was originally developed on, this work was carried out to test the iterative lysis methodology on a novel strain. In addition to helping to validate this methodology by applying it to a new strain, it was further optimized by improving the bead-based clean-up steps [16]. Sequencing this xenic lab-grown culture afforded not just the cyanobacterial genome but the genomes of six strains of heterotrophic bacteria derived from the metagenomic data acquired from the cyanobacterial-heterotroph assemblage.

## 2. Materials and Methods

### 2.1. Strain and Harvesting

#### 2.1.1. Strain Isolation and Growth

Strain BL-A-41 was originally isolated from water collected from Arkabutla Lake in Northern Mississippi [21]; it has been in continuous cultivation in the Boudreau Lab since then. These continuous cultures were grown in 50 mL of freshwater BG-11 media (following UTEX Culture Collection of Algae protocol but with stocks #5 and #6 combined into one stock) [22] in 125 mL Erlenmeyer flasks under grow lights (ca. 1550 lux on a 16:8 h light/dark cycle).

#### 2.1.2. Cell Harvesting

Continuous cultures were harvested after ca. two months of growth by transferring the culture to 50 mL Falcon tubes then centrifuging at 10,000× *g* at 13 °C for 10 min before discarding the supernatant by decanting and collecting the cell pellet. The cell pellet was frozen at −20 °C or −70 °C and stored for later use.

### 2.2. DNA Extraction, Size Selection, and Sequencing

#### 2.2.1. Single DNA Extraction

Two frozen cell pellets (both from 50 mL cultures as discussed above) were lyophilized overnight and then combined into a single microcentrifuge tube. To this, we added 1000 μL of P1 Buffer from the Omega Bio-Tek (Norcross, GA, USA) Plant DNA kit, as well as a scoop of glass beads (ca. 10–20 mg), 10 μL of lysozyme, and 10 μL of Proteinase K (beads, lysozyme at 50 mg/mL, and Proteinase K at 20 mg/mL from the Omega Bio-Tek E.Z.N.A. Bacterial DNA kit), and the mixture was vortexed briefly. The mixture was then heated on a 65 °C block for one hour, and every 20 min, the tube was inverted six times to mix. After lysis, the sample was centrifuged at 21,000× *g* at 13 °C for 2.0 min, and the supernatant was collected in a fresh tube. A total of 140 μL of P2 Buffer (Omega Bio-Tek Plant DNA kit) was added to the supernatant collection, and the tube was inverted six times to mix. This mixture was then centrifuged again to remove precipitates at 21,000× *g* at 13 °C for 10 min. The supernatant was collected into a fresh tube, 80% *v*/*v* of LCMS-grade isopropyl alcohol was added, and the tube was inverted six times to mix. The alcohol-precipitated DNA was collected by centrifugation at 21,000× *g* at 13 °C for 5.0 min. Once the liquid alcohol mixture was discarded, the pellet was treated with 300 μL of 65 °C purified water, and the sample was placed back on the 65 °C block for 10 min. To the resolubilized DNA, 4.0 μL of RNase A (at 10 mg/mL from Omega Bio-Tek) was added, then 150 μL of P3 Buffer (Omega Bio-Tek Plant DNA kit) and 300 μL of ethanol (200 proof), inverting six times to mix after each addition. This DNA sample was purified with the column from the Omega Bio-Tek Plant DNA kit using the manufacturer’s protocol, with the DNA being eluted with 65 °C elution buffer (Omega Bio-Tek) using first 75.0 μL incubated for 5.0 min at room temperature and then an elution of 50.0 μL incubated for 5.0 min at room temperature (with both elutions collected into the same microcentrifuge tube by centrifugation for 1.0 min at 21,000× *g* at 13 °C). A Qubit fluorescence quantification using the ×1 ds DNA kit showed sufficient material (at 106 ng/μL) to proceed with bead-based size selection.

#### 2.2.2. Iterative DNA Extraction

Two frozen cell pellets (both from 50 mL cultures as discussed above) were lyophilized overnight and then combined into one 50 mL Falcon tube. To this, we added 1000 μL of P1 Buffer from the Omega Bio-Tek Plant DNA kit, and the tube was vortexed briefly. The tube was warmed to 60–65 °C in a water bath for 10 min, and then the mixture was transferred to a microcentrifuge tube and centrifuged at 21,000× *g* at 13 °C for 2.0 min. Both the supernatant and pellet were collected separately. To the pellet, 800 μL of P1 Buffer was added as well as a scoop of glass beads, 10 μL of lysozyme, and 10 μL of Proteinase K (beads, lysozyme, and Proteinase K from the Omega Bio-Tek E.Z.N.A. Bacterial DNA kit), and the mixture was vortexed briefly. The mixture was then heated on a 65 °C block for 20 min. Both the supernatant and pellet were collected separately as above. To the pellet, again, 800 μL of P1 Buffer was added, as well as 10 μL of lysozyme and 10 μL of Proteinase K (lysozyme, and Proteinase K from the Omega Bio-Tek E.Z.N.A. Bacterial DNA kit), and the mixture was vortexed briefly. This mixture was heated on the 65 °C block for 1 h, inverting to mix six times at 20 and 40 min and then collecting the final supernatant by centrifugation as above. The three separate supernatants were processed as with the single DNA extraction protocol (above). A Qubit fluorescence quantification using the ×1 ds DNA kit showed sufficient material in each fraction (at 313, 109, and 190 ng/μL for fractions A–C, respectively) to proceed with bead-based size selection.

#### 2.2.3. Bead-Based Size Selection

To the DNA sample, we added one-half the volume of Sera-Mag (Cytiva) beads; this mixture was flicked for ≤1 min to mix (15 s only when repeating the protocol, see below) and then spun down briefly to collect the mixture in the bottom of the microcentrifuge tube. The mixture was let stand at room temperature for 7.0 min; then, it was placed on a magnetic rack and allowed to settle undisturbed for 5.0 min. The supernatant was removed carefully with a pipette so as to not disturb the beads and not leave drops of liquid behind. The beads were then washed with 2 × 200 μL of freshly prepared 85% ethanol, after which the beads were allowed to air dry for 7.0 min and taken off of the magnetic rack. A total of 60 μL of TE buffer was added, and the tube was flicked for 15 s to suspend the beads, then being spun down and let stand at room temperature for 7.0 min as before. Again, the tube was placed on the magnetic rack for 5.0 min, and this time the supernatant was collected with a wide-bore pipette tip into a fresh tube. This process was repeated with 30 μL of beads and an elution of 50 μL of TE and then again with 25 μL of beads and an elution of 42 μL of elution buffer (Omega Bio-Tek). The final DNA concentrations of these purified DNA extracts were measured on the Qubit (138 ng/μL for the single extraction and 143, 82.2, and 79.8 ng/μL for fractions A-C, respectively) as with the extracts, and the samples were diluted with elution buffer to the target concentration for the sequencing vendor (50 ng/μL).

#### 2.2.4. Nanopore Whole Genome Sequencing

The DNA samples were sequenced with a commercial vendor (Plasmidsaurus, Louisville, KY, USA, using their “Big Bacterial Genome” sequencing service). The vendor used V14 library prep chemistry for R10.4.1 cells on a PromethION (Oxford Nanopore Technologies, Oxford, UK) with Dorado basecalling on super-accuracy mode to produce the raw files processed below.

#### 2.2.5. Nanopore Targeted Amplicon Sequencing

An initial ribosomal sequence-based identification was achieved from an amplicon produced with primers targeting the cyanobacterial 16S or 23S genes (Appendix A) [23,24]. For closing of the cyanobacterial genome, primers were paired to span across the ribosomal repeats from the end of the linear cyanobacterial contigs in the initial draft assembly or paired from one end to within the ribosomal region where the longer amplicon could not be produced (Appendix A). These reactions used template DNA from the iterative DNA preparation (bead purified material from fraction B or C) with primers synthesized by Millipore Sigma (The Woodlands, TX, USA) using Q5 High-Fidelity 2× Master Mix polymerase (New England Biolabs, Ipswich, MA, USA); this polymerase was chosen for its high fidelity to aid in sequencing accuracy. For the thermal cycler method, see Appendix A. The ribosomal amplicon was observed to be a single clean band at the target length by gel electrophoresis of a small aliquot of the sample, so the remainder was purified using the DNA Clean & Concentrator-5 Kit (Zymo Research, Irvine, CA, USA), which was sufficient, as the gel imaging showed little to no off-target amplification. This column-purified ribosomal amplicon was then sequenced directly. The targeted amplicons for closing the genome were purified by gel electrophoresis in 1% TAE agarose (Sigma, Saint Louis, MO, USA, BioReagent for molecular biology, low EEO) ethidium bromide-stained gels run at 100 V for ca. 30 min and then recovered using the Zymoclean Gel DNA Recovery Kit (Zymo Research) to remove off-target amplicons. This DNA was below the target concentration for direct sequencing, so the purified band was reamplified as above, using it as the template in a fresh reaction, the product of which was purified with the Clean and Concentrator 5 Kit and sent to the vendor for nanopore sequencing (Plasmidsaurus, “Linear/PCR Standard Purified” service).

### 2.3. Bioinformatic Analyses

#### 2.3.1. Metagenomic Assembly

Raw genomic reads from the vendor were processed with the FiltLong tool (version 0.2.1). First, the reads were trimmed to a minimum length of 500 bp and then to a by-quality score to keep the top 99% of reads (or only filtering by length when after that step the remaining reads already fall below the 99% cutoff). Separately, all the raw read files were then reprocessed with FiltLong, with cutoffs of 1000 bp and 90%. Read datasets were analyzed with version 4.0.1 of the n50 tool to determine N50 statistics (see Appendix A). Flye (version 2.9.5-b1801) was used to assemble the metagenome using the input of the 500 bp/99% filtered reads pooled from all the separate fractions/preparations of DNA, with the nanopore high-quality preset and in metagenomics mode (see Appendix A) [25]. This assembly led to 761 fragments, including one circular contig of 4.63 Mb, with the N50 of these contigs being 3.28 Mb (N50 is a statistic that gives the length of the shortest contig that needs to be included to cover 50% of the dataset, with longer N50 values indicating a higher-quality dataset of longer reads/contigs). Using Bandage (version 0.8.1) showed that the largest 4.85 Mb contig was linear because of a bubble around a repeated region of ca. 5 kb that prevented the assembly of this contig with a smaller 145 kb linear fragment. In Geneious (version 2023.0.4), these two linear fragments were de novo assembled with the eight targeted amplicons of the *Pseudanabaena* sp. BL-A-41 ribosomal region using the Geneious assembler with the sensitivity set to “Highest Sensitivity/Slow” without trimming and set not to merge variants with coverage over approximately 6, merging homopolymer variants and circularizing contigs of 2 or more sequences, if ends matched. This assembly led to a single circular contig for the cyanobacterial chromosome. The two Flye-derived linear contigs were replaced by this circular contig in the metagenomic assembly, and the revised metagenomic assembly was medaka (version 2.0.1) polished using the combined FiltLong processed read files at 1000 bp and 90% (not the 500 bp and 99% read files that had been used to generate the initial Flye assembly); the medaka settings were a batch size of 32 with the r1041_e82_400bps_bacterial_methylation configuration file. The resulting medaka-polished metagenomic assembly was sorted by length, and all short contigs below 100 kb were removed. The remaining contigs were given a preliminary annotation via the DFAST platform (including a completeness check via CheckM against the phylum cyanobacteria for BL-A-41 and against the domain bacteria for all heterotrophic strains) [26], and in Geneious a Megablast comparison of the 16S ribosomal genes observed on these contigs, or a few ca. 2 kb, regions selected from across the contigs without ribosomal genes. This identified contigs as coming from the cyanobacterial host *Pseudanabaena* sp. BL-A-41 (one circular chromosome and three circular plasmids) as well as six heterotrophic strains were dubbed *Acidovorax* sp. BL-A-41-H1 (fifteen linear contigs), *Hydrogenophaga* sp. BL-A-41-H2 (a single linear contig), *Lysobacter* sp. BL-A-41-H3 (a single circular chromosome), *Novosphingobium* sp. BL-A-41-H4 (three linear contigs), *Sediminicoccus* sp. BL-A-41-H5 (a single linear contig), and *Tabrizicola* sp. BL-A-41-H6 (a single linear contig). Genome statistics are detailed in the Appendix A.

The assignment of the plasmids within the cyanobacterial genome was initially based on their circular architecture and size below 1 Mb in length. However, using geNomad (version 1.11.0) [27] to analyze the medaka polished metagenome scored the 159 and 231 kb plasmids very highly (0.9831 and 0.9855 plasmid scores, respectively) but did not score the 252 kb circular contig as a plasmid. This discrepancy is notable but will require further investigation to clarify the true nature of this contig, it is referred to as a plasmid based on its size (<1 Mb) and architecture (a circular contig).

#### 2.3.2. Preliminary 16S-Based Identification

In Geneious Prime (version 2023.0.4), the ribosomal region amplicon from *Pseudanabaena* sp. BL-A-41 was searched against public sequences in GenBank by Megablast for the 64 top hits. Redundant hits and uncultured 16S sequences with poor pairwise identity were removed, and the sequences were aligned in Geneious with MUSCLE using the PPP algorithm. The partial reads significantly shorter than the others were removed to allow for the creation of a new select alignment of genes with longer overlap. This alignment was trimmed to the region of overlap, and the Geneious Tree Builder was used to generate a phylogenetic tree with the Jukes–Cantor genetic distance model, using neighbor joining as the tree building method, and resampling with bootstraps at 1000 replicates. Based on this preliminary phylogeny, a new tree was generated using the same protocol except that a genetically distant strain identified in the initial tree was used as an outgroup (with any particularly distant hits removed if they did not add to the analysis). In the final tree, 100,000 bootstrap replicates were used to produce the final phylogeny tree (Appendix A).

For the heterotrophic strains, 16S genes were extracted in Geneious from the preliminary DFAST [26]-derived annotation of the metagenome, and the phylogeny analysis was run as above with the *Pseudanabaena* sp. BL-A-41 amplicon but only covering the 16S gene. Note that for *Hydrogenophaga* sp. BL-A-41-H2, the DFAST annotation failed to recognize the 16S gene at the edge of the linear contig, but by using a Megablast search, a putative 16S sequence for BL-A-41-H2 could be annotated by comparison with its top Megablast hit, a highly similar *Hydrogenophaga* genome (CP170553), for the same analysis as with the DFAST-annotated sequences. All these trees are available as Appendix A.

#### 2.3.3. Mapping of Reads to Genomes

The 1000 bp/90% FiltLong processed read files from fractions A, B, and C and the single extraction were mapped separately to the final medaka-polished MAGs (including plasmids for *Pseudanabaena* sp. BL-A-41 and separately all linear contigs from *Acidovorax* sp. BL-A-41-H1 or *Novosphingobium* sp. BL-A-41-H5) with Geneious (version 2023.0.4) using the Geneious mapper with the sensitivity setting of “Medium Sensitivity/Fast” with no fine tuning or trimming.

#### 2.3.4. Genome Comparison with the OrthoANI Tool (OAT) [28]

The chromosome of *Pseudanabaena* sp. BL-A-41 was extracted and compared by OAT (version 0.93.1) [28] to all of the available chromosome or complete level assemblies within the genus *Pseudanabaena* that were available on GenBank. For this analysis, chromosomes were extracted and compared, excluding plasmid sequences. The OAT was set to calculate the OrthoANI value using two threads, with the original ANI calculated in both directions with two different values managed by averaging and the GGDC set to 2. Also included in the analysis were the chromosomes of two prior cyanobacterial chromosomes assembled using the iterative lysis protocol, with *Leptolyngbya* sp. BL-A-14 (CP166621.1) and *Limnothrix* sp. BL-A-16 (CP166615.1) as outgroups [16].

#### 2.3.5. Preliminary Analysis of Natural Product Biosynthetic Capacity with antiSMASH [20]

The *Pseudanabaena* sp. BL-A-41 genome was submitted to the web platform of antiSMASH (version 7.1.0) [20], with the bacterial settings using a relaxed detection strictness. This analysis was also completed on the heterotrophic genomes (combined into one fasta file) with the same settings.

## 3. Results

### 3.1. Iterative Fractionation Allowed for Better Recovery of Cyanobacterial DNA Versus a Single Extraction

The results of mapping the reads to the final genomes indicated that though the proportion of reads coming from the cyanobacterium did not increase with each fractionation step (compare 12% of reads in Fraction A to 22% in Fraction B and 14% of reads in Fractions C), and all iterative fractions had a far superior proportion of cyanobacterial reads compared to the 1.6% of reads observed in the single DNA extraction (Table 1, Appendix A). For some strains of heterotrophic bacteria, e.g., *Acidovorax* sp. BL-A-41-H1, *Hydrogenophaga* sp. BL-A-41-H2, or *Tabrizicola* sp. BL-A-41-H6, fractionation did see a continuous decrease in the proportion of reads coming from the heterotroph during the iterative DNA extraction, as was predicted in the initial report on this technique [16]. However, this was not true in all cases; additionally, a majority of reads in each preparation came from the two heterotrophs *Lysobacter* sp. BL-A-41-H3 and *Novosphingobium* sp. BL-A-41-H4. However, combining those two strains together shows that while 85% of the single DNA extraction reads came from these two heterotrophs, in the iterative fractions 58%, 51%, and 59% of reads in fractions A-C, respectively, came from these two strains. In preliminary work on *Leptolyngbya* sp. BL-A-14 or *Limnothrix* sp. BL-A-16, the majority of reads in fractions B and C came from the cyanobacterium [16]. The number of heterotrophic strains observed within the *Pseudanabaena* sp. BL-A-41 culture and the fact that cyanobacterial reads were never the majority of any iterative fraction suggests that this strain has a more complex microbial community than the two strains sequenced in the preliminary work. There were large differences in coverage across the genomes in the initial Flye assembly, comparing 111× for the cyanobacterium to 36×, 57×, 264×, 431×, 35×, and 97× for BL-A-H1 through BL-A-41-H6, respectively (Appendix A). However, these results cannot deconvolute to what extent these differences derive from different abundances within original culture or differences in efficiency of DNA recovery during the extractions. Comparison of the iterative fractions to the single lysis protocol supports the argument that the iterative approach is helping to target the cyanobacterial DNA versus the heterotrophic microbiome, and the fractionation within a complex microbiome observed by read mapping shows the iterative lysis protocol is differentiating between strains.

### 3.2. Bead-Based Size Selection Provided Sufficient Removal of Sheared DNA Fragments to Allow for Sequencing with a Commercial Vendor

In preliminary sequencing results on another xenic cyanobacterial culture, the use of repeated bead-based clean-ups was investigated, which showed a logarithmic improvement in total bases sequenced but a much more modest linear improvement in the N50 of the reads (unpublished results). This result suggested to us that while the shortest, most sheared DNA molecules might interfere with nanopore library preparation or lead to the loss of pores during sequencing, repeated bead-based clean-ups could remove them. This allows for good metrics for the total bases sequenced but does little to improve on the physical quality of the remaining DNA. As such, during the iterative DNA lysis, where the sample is handled aggressively and column-based DNA purification is used, it was expected that the N50 of the reads would be poor relative to high-molecular-weight DNA preparations. Indeed, the N50 of the raw reads combined from all genome sequencing runs (before FiltLong processing) was only 2387 bp (compared to an N50 value of 9215 bp achieved recently in the same lab but with a high-molecular-weight genomic DNA preparation from a heterotrophic strain [29]). Little difference in the N50 was noted between the reads from the single extraction versus the iterative fractions (see Appendix A).

With shorter reads, repeated regions can become a problem during assembly, and unsurprisingly, in initial efforts, a repeated ca. 5.3 kb region prevented the closing of the cyanobacterial chromosome. Based on ribosomal rRNA genes at the linear contig ends, this repeated region was assigned as a copy of the ribosomal region. To close this bubble, another sequencing run was ordered with DNA pooled from the remainder of fractions B and C (the fractions with the highest proportion of cyanobacterial DNA); these reads and those from the single lysis DNA preparation were added to the original iterative-lysis reads. From the combined dataset, a newly generated metagenomic assembly (Figure 1A) still showed a bubble in the cyanobacterial chromosome (Figure 1B). As an alternative approach, primers were designed based on the linear contigs ends to span the gaps and allow for assembly across this bubble. To aid in this analysis, these primers were also paired with cyanobacterial ribosomal region primers (see Appendix A) [23,24]. Geneious was able to de novo assemble the two linear cyanobacterial contigs and the sequenced amplicons into a single circular chromosome for the cyanobacterium. This chromosome was uploaded to GenBank with the accession number CP192607, with the *Pseudanabaena* sp. BL-A-41 plasmids given accession number CP192604-CP192606.

In this metagenomic workflow, the *Lysobacter* sp. BL-A-41-H3 chromosome, which has a single ribosomal region, was also assembled into a closed circular contig. While a small bubble (547 bp) was observed in the assembly graph of the *Tabrizicola* sp. BL-A-41-H6 chromosome, the small size of this bubble suggests that the linear contig for this genome is nearly complete. While the remaining heterotrophic genomes are more fragmented, each had contigs of megabase length, allowing the publication of long continuous portions of their genomes in their partially complete genomes. These genomes were uploaded to GenBank with the accession numbers JBPVJQ000000000 for *Acidovorax* sp. BL-A-41-H1, CP192603 for *Hydrogenophaga* sp. BL-A-41-H2, CP192602 for *Lysobacter* sp. BL-A-41-H3, JBPVJP000000000 for *Novosphingobium* sp. BL-A-41-H4, CP192601 for *Sediminicoccus* sp. BL-A-41-H5, and CP192600 for *Tabrizicola* sp. BL-A-41-H6.

### 3.3. Bioinformatic Analysis Supported the Assignment of BL-A-41 as Belonging to the Genus Pseudanabaena

A ribosomal region-based phylogenetic analysis supported the assignment of strain BL-A-41 as belonging to the genus *Pseudanabaena* (see Appendix A), with the amplicon sequence having a top Megablast hit to *Pseudanabaena yagii* NIES-4237 (98.5% pairwise identity). A similar assessment of extracted 16S genes from the metagenomic assembled chromosomes of strains BL-A-41-H1 through BL-A-41-H6 suggested that these bacteria belonged to the genera *Acidovorax*, *Hydrogenophaga*, *Lysobacter*, *Novosphingobium*, *Sediminicoccus*, and *Tabrizicola*, respectively (see Appendix A).

The separate analysis of the BL-A-41 chromosome compared to publicly available cyanobacterial chromosomes with the OAT also supported this strain’s assignment as a member of the genus *Pseudanabaena*. The similarity was not high enough to assert that BL-A-41 is within the same species as any of the compared strains, but this strain did clade with the *Pseudanabaena* spp. (Figure 2). Note that within the comparison dataset were the chromosomes of previously sequenced strains from the Boudreau Lab collection (in the closely related genus *Limnothrix*, a fellow member of the Pseudanabaenaceae family, and the more distantly related genus *Leptolyngbya*). This analysis was consistent with the prior ribosomal region-based method.

Preliminary analysis of the biosynthetic capacity of this assemblage by antiSMASH [20] did not find many known clusters, but there were numerous clusters that may be worthy of future investigation based on their apparent novelty. This analysis showed four potential clusters on the cyanobacterial chromosome, one NRPS-like, one Type 3 PKS, one RRE-containing cluster, and one terpene, but none with similarity to known clusters. Within the heterotrophic genomes, the only hits with high similarity to known clusters were three terpene pathways found on the genomes of *Hydrogenophaga* sp. BL-A-41-H2, *Novosphingobium* sp. BL-A-41-H4, and *Tabrizicola* sp. BL-A-41-H6 (one cluster found in each genome). These pathways appear to make carotenoids, as assessed by the most similar pathway within known clusters. However, more distinct clusters were found across these genomes, with antiSMASH identifying 35 potential clusters total that could be explored in future work. Because few of the observed clusters had high homology to known pathways, it is hard to predict ecological roles or biotechnological significance of their corresponding natural products. It is known that siderophores from one producing organism can be utilized across microbial communities by non-producing strains [30,31], and the metal-binding domains of siderophores are well known and predictable from biosynthetic gene cluster sequences [20,31]. Unfortunately, siderophore pathways were not observed in the antiSMASH analysis, so proposed roles, such as aiding in host iron acquisition, cannot be proposed for the heterotrophic microbiome’s natural products at this time.

## 4. Discussion

In attempting to show that the iterative lysis method could target DNA from a cyanobacterium in a xenic culture, the initial results were disappointing because the chromosome was not assembled as a circular contig, and the fractionation did not lead to a continued increase in proportion of cyanobacterial reads with each fraction. However, after preparing an extraction with a traditional single lysis step showed only 1.6% of reads in that sample mapped to the cyanobacterial genome, the value of the iterative preparation in targeting the cyanobacterium in a xenic culture was made clear. Furthermore, even in a complex community, as in this culture, this metagenomic sequencing approach can lead to megabase-length assemblies for the heterotrophic strains as well the targeted cyanobacterium. The ability to fractionate DNA with iterative lysis steps may be applicable to other microbial communities, such as the microbial community within sponges [32,33,34], symbioses between fungi and photoautotrophs in lichens [35,36], or in environmental water or soil samples where environmental DNA sequencing has shown diverse microbial communities [37,38,39], though these conjectured broader microbiology applications will require independent validation.

The similar N50 results between the single extraction reads and the iterative fractions may indicate that the shared column purification step is driving much of the DNA shearing. In future work, other DNA isolation techniques can be explored to avoid this shearing. Though this is more shearing than is desirable for nanopore-based sequencing, bead-based clean-up steps did afford a DNA sample compatible with nanopore sequencing. Even before the use of targeted amplicons to close the circular genome, the metagenomic assembly led to a 96% complete contig for the cyanobacterial chromosome along with megabase-length contigs for the heterotrophic strains. With repeat regions being common in cyanobacterial genomes [11,40], similar issues may persist going forward. However, the near-complete draft BL-A-41 before closing of the gaps by amplicon sequence was complete enough to allow for antiSMASH annotation of all the natural product biosynthetic gene clusters annotated in the final genome. So, there is clear utility from this workflow for natural product chemistry investigations. While the assembly of the cyanobacterial genome allowed for genome-based comparison to show that this strain is likely novel.

Future targets for this work will be to apply this iterative lysis method to other xenic cultures of cyanobacteria to demonstrate its utility across the broad morphological diversity of cyanobacteria [41]. One limitation in this area might be that the iterative lysis method has been developed on filamentous cyanobacteria, and single-celled cyanobacteria may not see the same benefit to using the iterative lysis approach. As such, single-celled cyanobacteria, such as *Synechococcus* spp. [42,43], are an important clade to examine in future work. A potential improvement of the method to explore is using washing of the cell pellet to physically separate the cyanobacterial assemblage from the heterotrophic strains before the cell lysis step, as the washed pellet could better target the cyanobacterium, and cells collected from the washing could be sequenced to improve coverage of the heterotrophic microbiome.

## Figures and Tables

**Figure 1 microorganisms-13-01996-f001:**
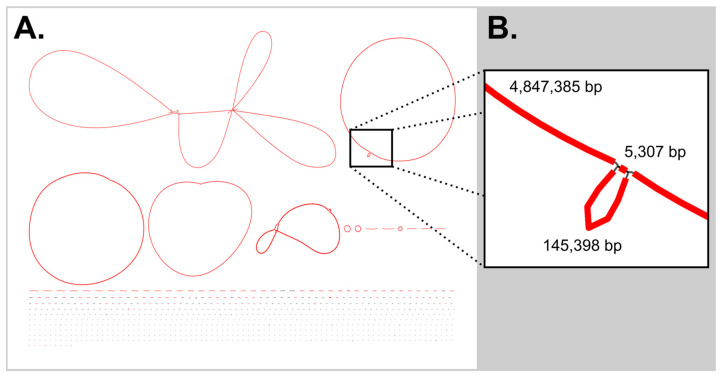
Bubbles in the Flye metagenomic assembly. (**A**) The Bandage (version 0.8.1) image of the initial Flye [25]-derived assembly, showed a bubble in the chromosome of the cyanobacterial genome, colored by depth of coverage. (**B**) This repeat region of 5307 bp is a copy of the cyanobacterial ribosomal region. Note: The *Lysobacter* sp. BL-A-41-H3 chromosome was circular, while the bubble in the chromosome of *Tabrizicola* sp. BL-A-41-H6 is too small to see in in the figure at this scale.

**Figure 2 microorganisms-13-01996-f002:**
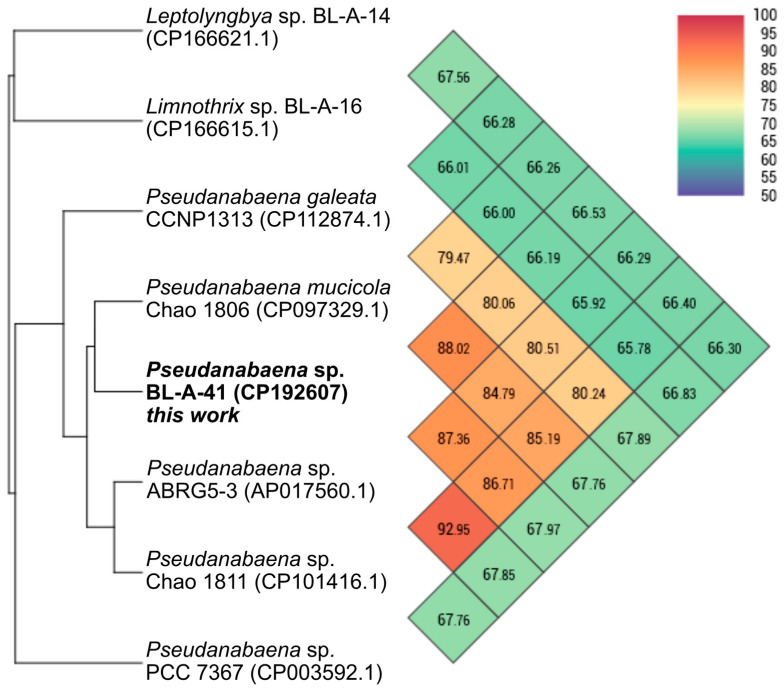
OrthoANI heatmap comparison of the *Pseudanabaena* sp. BL-A-41 chromosome. BL-A-41 chromosome (bold label) claded with the other *Pseudanabaena* chromosomes. This analysis was performed with the OAT [28].

**Table 1 microorganisms-13-01996-t001:** Mapping of 1000 bp/90% filtered reads to final genomes.

Genome Target for Mapping	Single DNA Extraction Reads	Iterative DNA Extraction Reads		
		Fraction A	Fraction B	Fraction C
Total Reads	130,032	297,976	426,824	384,944
*Pseudanabaena* sp. BL-A-41	2142 (1.6%)	36,339 (12%)	95,890 (22%)	54,116 (14%)
*Acidovorax* sp. BL-A-41-H1	8769 (6.7%)	22,754 (7.6%)	27,850 (6.5%)	22,252 (5.6%)
*Hydrogenophaga* sp. BL-A-41-H2	4422 (3.4%)	24,185 (8.1%)	26,874 (6.3%)	20,161 (5.2%)
*Lysobacter* sp. BL-A-41-H3	53,670 (41%)	77,723 (26%)	98,430 (23%)	117,471 (31%)
*Novosphingobium* sp. BL-A-41-H4	57,540 (44%)	94,858 (32%)	117,844 (28%)	109,502 (28%)
*Sediminicoccus* sp. BL-A-41-H5	1818 (1.4%)	8406 (2.8%)	15,457 (3.6%)	24,481 (6.4%)
*Tabrizicola* sp. BL-A-41-H6	2167 (1.7%)	33,090 (11%)	42,264 (9.9%)	34,723 (9.0%)

## Data Availability

The GenBank records for this work are available under BioProject number PRJNA1241331. The raw fastq read files from the genome sequencing runs and amplicon sequencing efforts have been uploaded into a sequence archive under the accession numbers SRR32868210 through SRR32868215. The final metagenome assembled genomes are available under the following accession numbers: CP192604-CP192607 for *Pseudanabaena* sp. BL-A-41, JBPVJQ000000000 for *Acidovorax* sp. BL-A-41-H1, CP192603 for *Hydrogenophaga* sp. BL-A-41-H2, CP192602 for *Lysobacter* sp. BL-A-41-H3, JBPVJP000000000 for *Novosphingobium* sp. BL-A-41-H4, CP192601 for *Sediminicoccus* sp. BL-A-41-H5, and CP192600 for *Tabrizicola* sp. BL-A-41-H6.

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
