# Peer review of "Metagenomic Assembled Genomes of a Pseudanabaena Cyanobacterium and Six Heterotrophic Strains from a Xenic Culture"

_microorganisms, 2025, doi:10.3390/microorganisms13091996_

Round 1

Reviewer 1 Report

Comments and Suggestions for Authors

The manuscript by Paul D. Boudreau presents a metagenomic analysis of a microbial consortium using an innovative iterative DNA fractionation method that improved cyanobacterial genome recovery. The authors successfully assembled complete genomes and provide rigorous phylogenetic characterization of this complex cyanobacterial-heterotroph community. The technical execution is exemplary with careful resolution of assembly challenges, though the ecological significance of the community structure warrants deeper discussion.

I recommend accept with minor revision - this represents solid methodological innovation coupled with quality biological discovery.

  1. Address the conflicting plasmid classification results between geNomad scoring and initial assignment.
  2. Present key metrics like genome coverage depth and read mapping rates in the main text rather than supplementary materials.
  3. Discuss potential metabolic interactions and ecological roles within this cyanobacterial-heterotroph consortium.
  4. Provide more detailed discussion of the ecological or biotechnological significance of the identified novel biosynthetic pathways.
  5. Acknowledge when the iterative fractionation approach might not be suitable for certain community types or research questions.

Author Response

The manuscript by Paul D. Boudreau presents a metagenomic analysis of a microbial consortium using an innovative iterative DNA fractionation method that improved cyanobacterial genome recovery. The authors successfully assembled complete genomes and provide rigorous phylogenetic characterization of this complex cyanobacterial-heterotroph community. The technical execution is exemplary with careful resolution of assembly challenges, though the ecological significance of the community structure warrants deeper discussion.

I recommend accept with minor revision - this represents solid methodological innovation coupled with quality biological discovery.

1. Address the conflicting plasmid classification results between geNomad scoring and initial assignment.

Though we have not resolved this conflict, we have expanded the discussion to mention that geNomad relies on a known database of marker genes, so a truly novel plasmid, may score poorly with that tool. An explanation which does make sense with our work to sequence a novel organism.

2. Present key metrics like genome coverage depth and read mapping rates in the main text rather than supplementary materials.

Mapping rates from SI Table S3 has been moved to the main text as Table 1, depth of coverage has been mentioned in the Results as well.

3. Discuss potential metabolic interactions and ecological roles within this cyanobacterial-heterotroph consortium.

An expanded discussion of the potential exchange of natural products, as in iron acquisition by siderophores, including citations of prior reports in this area, has been added to the Discussion (see response to next comment).

4. Provide more detailed discussion of the ecological or biotechnological significance of the identified novel biosynthetic pathways.

This is difficult because of the few known clusters detected in the analysis, so much more discussion would be highly speculative. However, the following text has been added to the discussion to detail other potential areas where the fractionation approach might be applied:

The ability to fractionate DNA with iterative lysis steps may be applicable to other microbial communities, such as the microbial community within sponges,32–34 symbioses between fungi and photoautotrophs in lichens,35,36 or in environmental water or soil samples where environmental DNA sequencing has shown diverse microbial communities.37–39 Though these conjectured broader microbiology applications will require independent validation.

5. Acknowledge when the iterative fractionation approach might not be suitable for certain community types or research questions.

Mention of the fact that this method was developed with filamentous cyanobacteria, and has yet to be validated on single-celled cyanobacteria, which may respond differently to the iterative lysis, has been added to the discussion.

Reviewer 2 Report

Comments and Suggestions for Authors

In the reviewed MS, the results of the new-to-science and very useful for cyanobacterial research are presented. Obtaining axenic cultures of cyanobacteria is a very difficult task. In this regard, the new protocols for the identification of cyanobacteria and bacteria from their association are important to microbiology. The author modified previously described methods and was able to achieve valuable results. The Methods section is very detailed. It is necessary to note that the data is illustrated by clear figures. In supplementary materials, phylogenetic trees, primer lists, and other useful information are presented.

I read the paper with great interest. I think that the MS should be published in the journal Microorganism after some corrections.

 Major suggestions:

  1. Correct the MS according to the journal template (reference and figure citation styles; dots in section titles; unnecessary text underlining in the description of methods; reference number style).
  2. Add several sentences with the definition of cyanobacteria at the beginning of the introduction.
  3. In conclusion, please add a paragraph about the importance of your study to microbiology.
  4. Please transfer sequences accession numbers to the Results section.
  5. Add several recent (published in 2020-2025) papers to the reference list.

Minor suggestions:

  1. Line 3: Delete the dots in the title.
  2. Lines 10, 62: Try to avoid first-person narration.
  3. Line 22: Replace the words "cyanobacteria" and "metagenomics" with other terms, since you used very similar words in the title. Add another keyword, for example, DNA, cyanobacterial chromosome, Acidovorax, Hydrogenophaga, Lysobacter, Novosphingobium, Sediminicoccus, or Tabrizicola.
  4. Lines 265-273: This part of the MS is better to move to the Results.
  5. Line 277: In the beginning of the Results section, please add something like “The results of the investigation indicated that though the proportion of reads…”
  6. Lines 281-294: Latin names of genera should be italicized.
  7. Lines 298-300: This sentence is too complex; please correct that.

Author Response

In the reviewed MS, the results of the new-to-science and very useful for cyanobacterial research are presented. Obtaining axenic cultures of cyanobacteria is a very difficult task. In this regard, the new protocols for the identification of cyanobacteria and bacteria from their association are important to microbiology. The author modified previously described methods and was able to achieve valuable results. The Methods section is very detailed. It is necessary to note that the data is illustrated by clear figures. In supplementary materials, phylogenetic trees, primer lists, and other useful information are presented.

I read the paper with great interest. I think that the MS should be published in the journal Microorganism after some corrections.

 Major suggestions:

1. Correct the MS according to the journal template (reference and figure citation styles; dots in section titles; unnecessary text underlining in the description of methods; reference number style).

The subsubsections have been revised according to the journal style and the periods have been removed from the subsection titles. We have also revised the figure citation so it is not bolded.

2. Add several sentences with the definition of cyanobacteria at the beginning of the introduction.

The introduction of cyanobacteria and their history have been expanded at the start of the introduction.

3. In conclusion, please add a paragraph about the importance of your study to microbiology.

To provide broader context, other examples of microbial communities with divergent cell morphology and great research interest, such as the sponge microbiome, have been mentioned in the conclusion. There the iterative lysis approach might have applications outside my lab’s interest in cyanobacteria.

4. Please transfer sequences accession numbers to the Results section.

Accession numbers have been added to the results section.

5. Add several recent (published in 2020-2025) papers to the reference list.

Several references have been added including the following recent publications:

  • My lab’s genome announcement from 2025 to provide comparison of N50’s in this work versus a high molecular weight prep.
  • Hirose, 2021: to support the need for cyanobacterial genomics to resolve cyanobacterial phylogenies.
  • Khalifa, 2021: to highlight natural product value from cyanobacteria.
  • Sánchez-Baracaldo, 2022: a background paper on cyanobacterial roles in biogeochemical cycles.
  • Lopez-Igual, 2022 and Kono 2022: on the Synechococcus which are mentioned as a future target.

Minor suggestions:

1. Line 3: Delete the dots in the title.

Edited accordingly.

2. Lines 10, 62: Try to avoid first-person narration.

Edited accordingly.

3. Line 22: Replace the words "cyanobacteria" and "metagenomics" with other terms, since you used very similar words in the title. Add another keyword, for example, DNA, cyanobacterial chromosome, Acidovorax, Hydrogenophaga, Lysobacter, Novosphingobium, Sediminicoccus, or Tabrizicola.

Replaced with “DNA extraction” and “cell lysis”.

4. Lines 265-273: This part of the MS is better to move to the Results.

Lines 265-270 were maintained in the methods, with their discussion of the antiSMASH settings, while lines 270-273 were moved to the Results.

5. Line 277: In the beginning of the Results section, please add something like “The results of the investigation indicated that though the proportion of reads…”

Edited accordingly.

6. Lines 281-294: Latin names of genera should be italicized.

Edited accordingly.

7. Lines 298-300: This sentence is too complex; please correct that.

This section has been revised.

Reviewer 3 Report

Comments and Suggestions for Authors

This manuscript describes a metagenomic assembly pipeline aiming to rescue the genome of a Pseudanabaena cyanobacterium and six co-cultured heterotrophic bacterial strains from a xenic culture through a DNA lysis protocol refinement. The author shows that this technique enriches the percentage of reads recovering Cyanobacteria over single lysis techniques and successfully retrieves a number of high-quality genomes. Please, find below the comments to help you to enhance your manuscript:

Major Comments

  1. Need strong hypothesis with clear research question: The manuscript is supposed to represent a defined hypothesis or central question. Emphasize and why the iterative lysis approach is important in the biological or methodological question being asked?
  2. High descriptive, low analytical: the manuscript explains how the study was done and what was found in detail but does not delve beyond the details to interpret or explain them. For instance, there’s no statistical comparison of DNA yields, assembly metrics, or read proportions across preparations. Please provide quantitative comparisons (with statistical support) where appropriate—especially for read proportions, genome completeness, and DNA quality.
  3. It is crucial to re-arrange and avoid repetition: The Materials and Methods section contains too much of a list of equipment and step by step instruction, those could be included in the SI. A number of findings and interpretations are simply stated or repeated almost verbatim elsewhere in the discussion. Refactor the text to make the paper clearer and make detailed procedures additional material.
  4. Limited discussion of limitations and broader impact: The Discussion section does not critically assess the limitations of the iterative lysis approach (e.g. DNA shearing, non-uniform read proportions). Broader applications are mentioned briefly without detail or context. Discuss how the method could be generalized, potential issues (such as, community variability, DNA integrity), and its value in microbial ecology or biotechnology.
  5. Inconsistent or Incomplete Data Presentation: Key information such as total number of reads, N50/L50 values, genome completeness, and contamination are missing or inconsistently reported. Please provide summary tables for genome assembly metrics and include completeness/contamination estimates (like, CheckM or BUSCO results).

Minor comments

L7, "can be a challenge" better as "is often challenging"

L11-12, Fragmented sentence. Consider merging: "...cyanobacterial chromosome; however, repeated ribosomal regions caused assembly issues."

L14, "saw far smaller proportion" to "yielded a markedly lower proportion"

L19-21, rewrite some part as "This study demonstrates that iterative lysis enriches for cyanobacterial DNA and enables concurrent genome assembly of cohabiting heterotrophs."

L25, rewrite as "Cyanobacteria are photosynthetic bacteria that have evolved large cell sizes..."

L34, Typo – "larges" should be "large"

L34-36, Long sentence. Please rewrite for the better expression

L39-44, The paragraph is overly descriptive. Condensing this with a clearer research gap.

L58-61, Better placed in Methods or a brief definition earlier in Intro.

L69-79, Equipment listing too detailed. Move to Supplementary if not essential for replication.

L93-98, Please consider summarizing DNA extraction protocol, very standard, overly detailed.

L117-134, The iterative DNA lysis steps are repeated multiple times. Move to Supplementary.

L154, Report number of reads per sample and yield per preparation.

L164-174, Provide justification for using specific kits and amplicon lengths.

L184-185, Clarify what N50 means and why it matters.

L193, Mention genome completeness/contamination if assessed (e.g., with CheckM or BUSCO).

L204-213, Too many stats in-text. Consider summarizing in a table.

L216-220, Clarify the contradiction in plasmid prediction why is 252 kb still considered a plasmid?

L277-280, Provide standard deviation or replicates if available.

L284-286, Consider giving bar plot or pie chart for each fraction’s read composition.

L306-308, Define "N50" earlier. Clarify what value is acceptable and why it’s suboptimal here.

L318-330, Very descriptive. Add figure labels (e.g., “see Figure 1A”) and simplify language.

L361-364, Repetitive. Avoid repeating data values. Focus on interpreting significance.

L367-369, Clarify whether the method is generalizable to other microbial cultures.

L387-391, This could be reframed as a future direction.

L408-418, Double-check GenBank IDs – placeholders should be updated before publication.

Author Response

This manuscript describes a metagenomic assembly pipeline aiming to rescue the genome of a Pseudanabaena cyanobacterium and six co-cultured heterotrophic bacterial strains from a xenic culture through a DNA lysis protocol refinement. The author shows that this technique enriches the percentage of reads recovering Cyanobacteria over single lysis techniques and successfully retrieves a number of high-quality genomes. Please, find below the comments to help you to enhance your manuscript:

Major Comments

1. Need strong hypothesis with clear research question: The manuscript is supposed to represent a defined hypothesis or central question. Emphasize and why the iterative lysis approach is important in the biological or methodological question being asked?

The Intro is revised to detail our hypothesis that the iterative lysis methodology is broadly applicable, and the opening of the Discussion is rewritten to more clearly tie to that hypothesis.

2. High descriptive, low analytical: the manuscript explains how the study was done and what was found in detail but does not delve beyond the details to interpret or explain them. For instance, there’s no statistical comparison of DNA yields, assembly metrics, or read proportions across preparations. Please provide quantitative comparisons (with statistical support) where appropriate—especially for read proportions, genome completeness, and DNA quality.

Genome completeness and contamination are now included in the SI, assembly metrics and read statistics are included. However, to save costs, the DNA extraction was only carried out once with each method, so statistical support cannot be assigned to the mapped read percentages.

3. It is crucial to re-arrange and avoid repetition: The Materials and Methods section contains too much of a list of equipment and step by step instruction, those could be included in the SI. A number of findings and interpretations are simply stated or repeated almost verbatim elsewhere in the discussion. Refactor the text to make the paper clearer and make detailed procedures additional material.

Equipment section of Materials and Methods moved to the SI. Repetitive mention of the bubble size in the Tabrizicola genome was removed from the Figure 1 legend and is now only mentioned in the Results.

4. Limited discussion of limitations and broader impact: The Discussion section does not critically assess the limitations of the iterative lysis approach (e.g. DNA shearing, non-uniform read proportions). Broader applications are mentioned briefly without detail or context. Discuss how the method could be generalized, potential issues (such as, community variability, DNA integrity), and its value in microbial ecology or biotechnology.

Discussion of broader impact and limitations of the method have been expanded as well as mention of the likely source of DNA shearing, the column purification step, is added. Note that this is a change from the original manuscript, but in accumulating the N50 values of all the reads for the reviewers, it was notable that there was little difference between the single and iterative read sets (see Table S3). So, the discussion has been changed to blame the shearing on the column purification step that is shared between the two different protocols.

5. Inconsistent or Incomplete Data Presentation: Key information such as total number of reads, N50/L50 values, genome completeness, and contamination are missing or inconsistently reported. Please provide summary tables for genome assembly metrics and include completeness/contamination estimates (like, CheckM or BUSCO results).

CheckM estimates for completeness from the DFAST annotations have now been included. Total reads from each fraction are now included in Table 1 (former Table S3 moved to the main text). In the SI the N50 of the reads included in the SI. The assembly summary for the original Flye metagenomic assembly is now included in the SI.

Minor comments

L7, "can be a challenge" better as "is often challenging"

Edited accordingly.

L11-12, Fragmented sentence. Consider merging: "...cyanobacterial chromosome; however, repeated ribosomal regions caused assembly issues."

Unfortunately, I don’t follow this suggestion, the sentence on line 11-12 isn’t split there’s only a comma between chromosome and however: “This effort led to the assembly of a megabase-length cyanobacterial chromosome, however, repeated ribosomal regions created assembly issues even after adding data from another sequencing run to improve coverage.”

L14, "saw far smaller proportion" to "yielded a markedly lower proportion"

Edited accordingly.

L19-21, rewrite some part as "This study demonstrates that iterative lysis enriches for cyanobacterial DNA and enables concurrent genome assembly of cohabiting heterotrophs."

Edited accordingly.

L25, rewrite as "Cyanobacteria are photosynthetic bacteria that have evolved large cell sizes..."

Edited accordingly.

L34, Typo – "larges" should be "large"

Edited accordingly.

L34-36, Long sentence. Please rewrite for the better expression

Edited accordingly.

L39-44, The paragraph is overly descriptive. Condensing this with a clearer research gap.

This paragraph has been condensed and more citations have been added to show the impact of cyanobacterial sequencing efforts (Hirose, 2021).

L58-61, Better placed in Methods or a brief definition earlier in Intro.

A brief summary was moved earlier in the Intro.

L69-79, Equipment listing too detailed. Move to Supplementary if not essential for replication.

Moved what was subsection 2.1 to the SI.

L93-98, Please consider summarizing DNA extraction protocol, very standard, overly detailed.

[See response to comment on L117-134, below].

L117-134, The iterative DNA lysis steps are repeated multiple times. Move to Supplementary.

I appreciate the desire for brevity, however, as the central focus of this work is validating a method for DNA extraction, I believe this detail is warranted. A key result of this work is the comparison between the single to iterative lysis (which wasn’t run in the original report to save on cost), so while, yes, the single and iterative lysis descriptions are repetitive. I’d argue the detail is necessary to document the difference between the protocols.

L154, Report number of reads per sample and yield per preparation.

Reads per sample are available in Table 1, yields from extracts listed in subsubsections 2.3.1 and 2.3.2, and concentrations after bead purification are now included in 2.3.3.

L164-174, Provide justification for using specific kits and amplicon lengths.

The kit justification has been added, as discussed in the text, primers were designed off the ends of the linear contigs, so amplicon length was not directly predictable, and why I used the long 8.0 min extension time given in Table S2.

L184-185, Clarify what N50 means and why it matters.

A definition of N50 is now provided.

L193, Mention genome completeness/contamination if assessed (e.g., with CheckM or BUSCO).

CheckM results from the DFAST annotations has been added to an SI Table.

L204-213, Too many stats in-text. Consider summarizing in a table.

The GC content and lengths have been moved to an SI Table.

L216-220, Clarify the contradiction in plasmid prediction why is 252 kb still considered a plasmid?

We don’t think it’s a contradiction that geNomad didn’t score this plasmid, as that tool searches against a database, if the plasmid is novel it may have few marker genes, and cyanobacteria are undersequenced. 252 kb is still a reasonable size to call it a plasmid, this contig lacks core housekeeping genes such as the rRNA genes, so assigning it as a secondary chromosome seems unwarranted.

L277-280, Provide standard deviation or replicates if available.

We do not have standard deviations as only one fractionation was run. This was mostly to save on sequencing cost, and as we were able to assemble the genome, we believe the results are still of interest.

L284-286, Consider giving bar plot or pie chart for each fraction’s read composition.

The bar chart is included in the Table of Contents Figure, it has also been added to the SI.

L306-308, Define "N50" earlier. Clarify what value is acceptable and why it’s suboptimal here.

A definition of N50 is now provided earlier with a comparison to a recent N50 for a HMW preparation in my lab for context.

L318-330, Very descriptive. Add figure labels (e.g., “see Figure 1A”) and simplify language.

Text has been simplified and references to 1A and 1B have been included.

L361-364, Repetitive. Avoid repeating data values. Focus on interpreting significance.

This text has been revised as advised by another reviewer, as such, it is no longer repeated.

L367-369, Clarify whether the method is generalizable to other microbial cultures.

A sentence has been added to highlight some other microbial communities which might be of interest.

L387-391, This could be reframed as a future direction.

I didn’t frame it as such as I’m trying to focus limited resources in the lab on further validating this iterative lysis methodology. Future directions have been expanded to include discussion of targeting diverse cell morphologies of host cyanos. 

L408-418, Double-check GenBank IDs – placeholders should be updated before publication.

The GenBank accession codes have been updated to remove placeholders.

Round 2

Reviewer 3 Report

Comments and Suggestions for Authors

Congratulations